

# Use and application of iNaturalist on land snails from Argentina

Ana Carolina Díaz[1,2] and Stella M. Martin[1]

[1] División Zoología Invertebrados, Facultad de Ciencias Naturales y Museo, Universidad Nacional de La Plata, La Plata, Buenos Aires, Argentina
[2] CONICET-Consejo Nacional de Investigaciones Científicas y Técnicas, La Plata, Buenos Aires, Argentina

## ABSTRACT

Gastropods are a large and diverse taxonomic group, and South America has an impressive diversity of land snails. However, there are no accurate and complete lists or estimates of native species' richness in South America. The aim of this work is to evaluate the use of iNaturalist in Argentina for terrestrial gastropods and its potential to contribute to the knowledge of malacofauna. A search was performed for Argentina on 15 June 2024, filtering observations for terrestrial gastropods, yielding 3,758 records, of which about 10% were of interest for this work. Exotic species represented between 60–63% of the observations in iNaturalist, and native species represented between 37–40%, with post-validation and pre-validation respectively. The geographical distribution of exotic species was mostly concentrated in the urban areas of the capital cities, and the observations of native species were concentrated in a few provinces. It was possible to detect and expand the distribution area of exotic species such as *Rumina decollata*, *Limacus flavus*, *Bradybaena similaris*, *Deroceras laeve*, *Deroceras reticulatum*, *Deroceras invadens*, *Arion intermedius*, *Milax gagates*, *Limax maximus*, *Vallonia pulchella*, possibly *Laevicaulis alte*; native species such as *Phyllocaulis soleiformis*, *Drymaeus poecilus*, *Drymaeus papyraceus*. Potential species not recorded in Argentina were identified as *Helix pomatia*, *Mesembrinus gereti*. We also recorded species within the known range, potential new species not described by science, a possible case of accidental transfer of *Mesembrinus interpunctus*, and the prediction of the distribution of *Megalobulimus lorentzianus* was verified. Through the development of this study, we were able to demonstrate the relevance of citizen science in providing interesting contributions to the knowledge of terrestrial mollusks biodiversity in Argentina.

# INTRODUCTION

Gastropods comprise 80% of mollusks, with an estimated 24,000–25,000 described species of terrestrial gastropods worldwide (*Lydeard et al., 2004*; *Rosenberg, 2014*). South America has a high diversity of land snails, many of which are endemic to the subcontinent or part of it (*Beltramino et al., 2015*). The global number of terrestrial mollusks has not yet been estimated, and information is scarce in some countries. Only two genera of Orthalicoidea are known from Guyana (*Breure, 2009a*). However, the knowledge of malacofauna is

Corresponding author
Ana Carolina Díaz,
anacdiaz@fcnym.unlp.edu.ar

progressing in most South American countries. In Suriname, *Breure (2009b)* reported 54 species; in Bolivia, *Zischka (1953)* catalogued more than 160 species of land snails; in Brazil, *Salvador et al. (2024)* counted 715 native species; Peru is one of the most species-rich countries in the Neotropics (*Breure & Mogollón Avila, 2010*) with 763 species (*Ramírez, Paredes & Arenas, 2003*); in Uruguay, *Scarabino (2003)* listed 42 native species/subspecies and 17 introduced species; *Breure, Ablett & Roosen (2022)* listed 219 species, including native and exotic, for Ecuador; in Chile *Araya, Miquel & Martínez (2017)* mention 160 species of native terrestrial gastropods. For Paraguay, *Quintana (1982)* catalogued 102 species between native and exotic terrestrial mollusks. In Colombia *Vera Ardila (2008)* listed 80 genera of terrestrial mollusks belonging to about 40 families. *Agudo-Padrón (2023)* counted 144 native and exotic species for Venezuela. In French Guiana *Gargominy & Muratov (2012)* mentioned 34 species and believe that there are many micromollusk species to be described. For Argentina, *Fernández (1973)* catalogued 269 species and subspecies, to which must be added the six described later (*Cuezzo, 2006*; *Miranda & Cuezzo, 2014*; *Salas Oroño, 2021*). Despite being a large group, gastropods remain a taxonomically poorly known class with many undescribed species (*Rosa, Cavallari & Salvador, 2022*), and there are no accurate and comprehensive lists or estimates of native species richness in South America (*Miyahira et al., 2022*).

In recent years, citizen science has gained momentum and is becoming more widespread around the world (*Bonney et al., 2014*). The literature has shown that engaging volunteers to participate in data collection is of great value to biodiversity knowledge on an unprecedented scale (*Bonney et al., 2014*). Citizen science platforms allow free and open access to a large amount and variety of open source biological data that would otherwise be impossible to collect due to the time and financial resources required (*Cohn, 2008*; *Braschler, 2009*; *Dickinson, Zuckerberg & Bonter, 2010*). With the information collected by volunteer users, it is possible to expand applied knowledge in conservation (*McKinley et al., 2017*), detect the timing and distribution of introduced species (*Vendetti et al., 2018*; *Rosa et al., 2022*), provide georeferenced data points (*Pimm et al., 2015*; *Bik, 2017*), and create accurate and comprehensive inventories (*Aravind, 2013*) of malacofauna on a national or regional scale. Additionally, it allows researchers to obtain records of rare or poorly known species, detect possible new species, discover the appearance of their soft parts, find individuals of species thought to be extinct, display ecological information (*Rosa, Cavallari & Salvador, 2022*), among many other applications. The aim of this article is to evaluate the use and application of the iNaturalist platform in Argentina for land snails, review the quality of the observations, and critically analyze the data of native species and the distribution of exotic species.

## MATERIALS AND METHODS

In the iNaturalist platform (https://www.inaturalist.org/observations), a search was performed by country, in this case Argentina, in which observations were filtered for terrestrial gastropods in the Subclass Heterobranchia, Superorder Eupulmonata.

The search was performed on June 15, 2024, where we distinguished observations registered by the *Community of iNaturalist (2024)* with pre-validation, which refers to

those observations that still require identification and community consensus to achieve accurate identification, and post-validation, that is, with "research grade" when the community agrees on an ID at the species level or below, *i.e.*, if at least two-thirds of the community identifiers validated it.

The data collected in the available general and specific bibliography, original descriptions of the species, and images of available type material were used to analyze the records. All observations for each species were reviewed with our user profiles to evaluate the accuracy of the identifications, as well as the localities of origin and current distribution in the country. Both native and exotic species were analyzed. When an incorrect previous identification was observed, it was updated. As the check progressed, those records of interest were copied and separated. The observations that were considered of interest were: those that extended the known range of the species, those that could be useful for the detection of exotic species not yet cited for Argentina, those that required collection for identification, those that confirmed the known range in the literature, those that could correspond to new species, misidentifications, cases of interaction with another species and the first images of the animal in life. The results obtained after compiling all these records of interest were then analyzed and discussed in depth. Some observations of interest are listed below under a number, this record is available online at the end of the observation URL: https://www.inaturalist.org/observations/ (*e.g.*, observation 215037922, available at https://www.inaturalist.org/observations/215037922).

## RESULTS

The terrestrial gastropod search in iNaturalist for Argentina as of June 15, 2024, yielded a total of 3,308 observations for 74 taxa, including both native and exotic representatives. Exotic species accounted for between 60–63% (with post-validation and pre-validation respectively), of the observations in iNaturalist. The species with the highest number of records were *Cornu aspersum* (O. F. Müller, 1774), *Rumina decollata* (Linnaeus, 1758), *Limacus flavus* (Linnaeus, 1758), *Otala punctata* (O. F. Müller, 1774), *Theba pisana* (O. F. Müller, 1774), *Lissachatina fulica* (Bowdich, 1822), *Bradybaena similaris* (A. Férussac, 1822), *Ambigolimax valentianus* (A. Férussac, 1821), *Meghimatium pictum* (Stoliczka, 1873), *Deroceras reticulatum* (O. F. Müller, 1774), *Limax maximus* Linnaeus, 1758 (Fig. 1A). For native species, the percentage of observations ranged from 37% to 40%, with the species with the highest number of records being *Bulimulus bonariensis* (Rafinesque, 1833), *Phyllocaulis soleiformis* (d'Orbigny, 1835), *Drymaeus poecilus* (d'Orbigny, 1835), *Megalobulimus lorentzianus* (Doering, 1876), *Megalobulimus sanctaepauli* (Ihering & Pilsbry, 1900), *Plagiodontes patagonicus* (d'Orbigny, 1835), *Drymaeus papyraceus* (Mawe, 1823), *Mesembrinus interpunctus* (Martens, 1886), *Phyllocaulis variegatus* (Semper, 1885), *Epiphragmophora trenquelleonis* (Pfeiffer, 1850) (Fig. 1B).

Figures 2A and 2C show the general geographic distribution of the observations in iNaturalist of exotic and native species in Argentina, while Figs. 2B and 2D show the geographic distribution of each species recorded in iNaturalist, exotic and native, respectively. From the total number of records (Data Set 1), a selection was made of those that were of interest. Records were selected mainly because of the geographic distribution

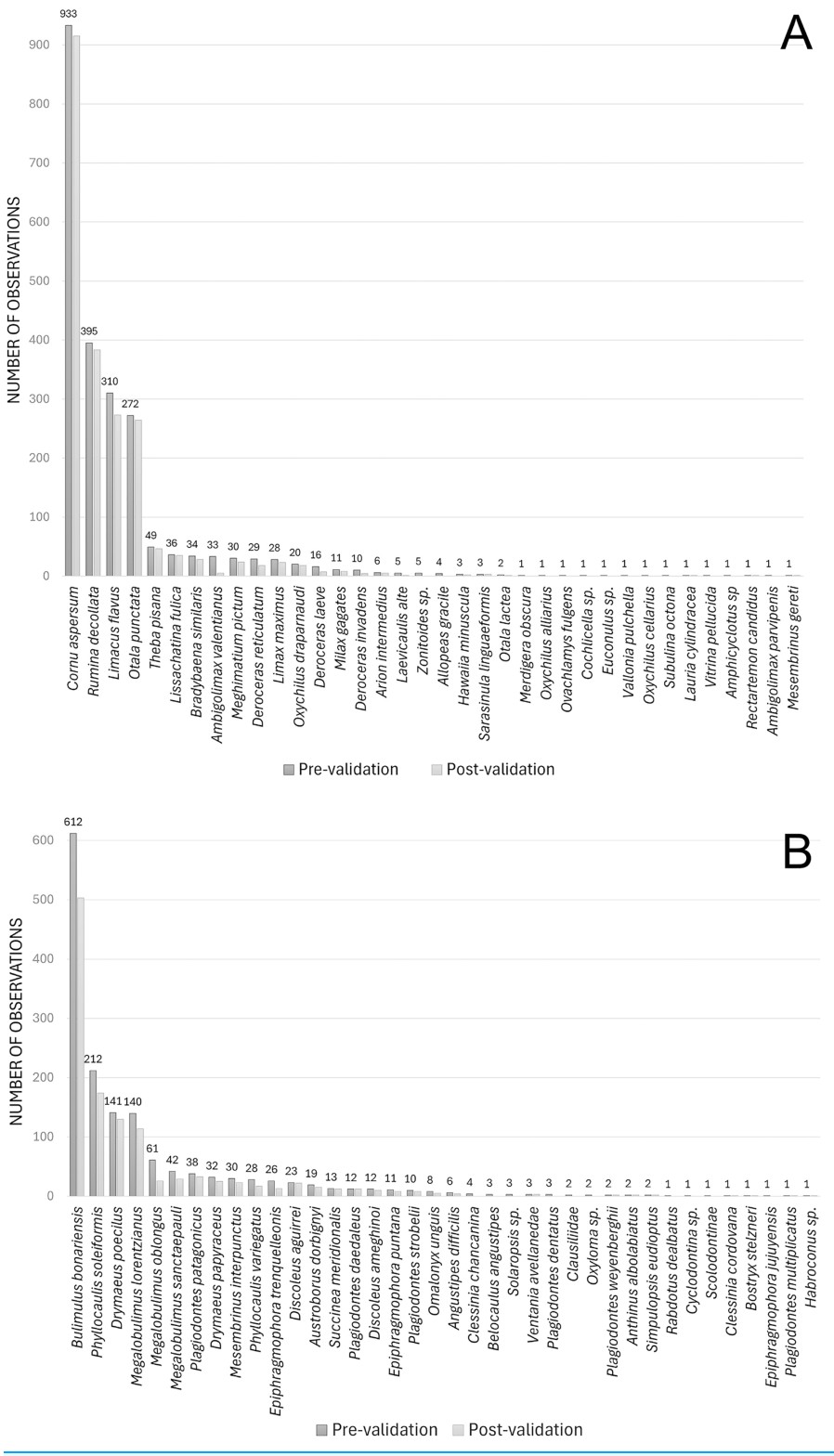

**Figure 1 Observations of terrestrial gastropods published in iNaturalist for Argentina as of June 15, 2024.** Number of observations with pre-validation (observations that still require identification and community consensus) and observations with post-validation (when the community agrees with an identification, *i.e.*, if at least two thirds of the community identifiers have validated it) for exotic (A) and native (B) species recorded. Image credit: Díaz Ana Carolina.

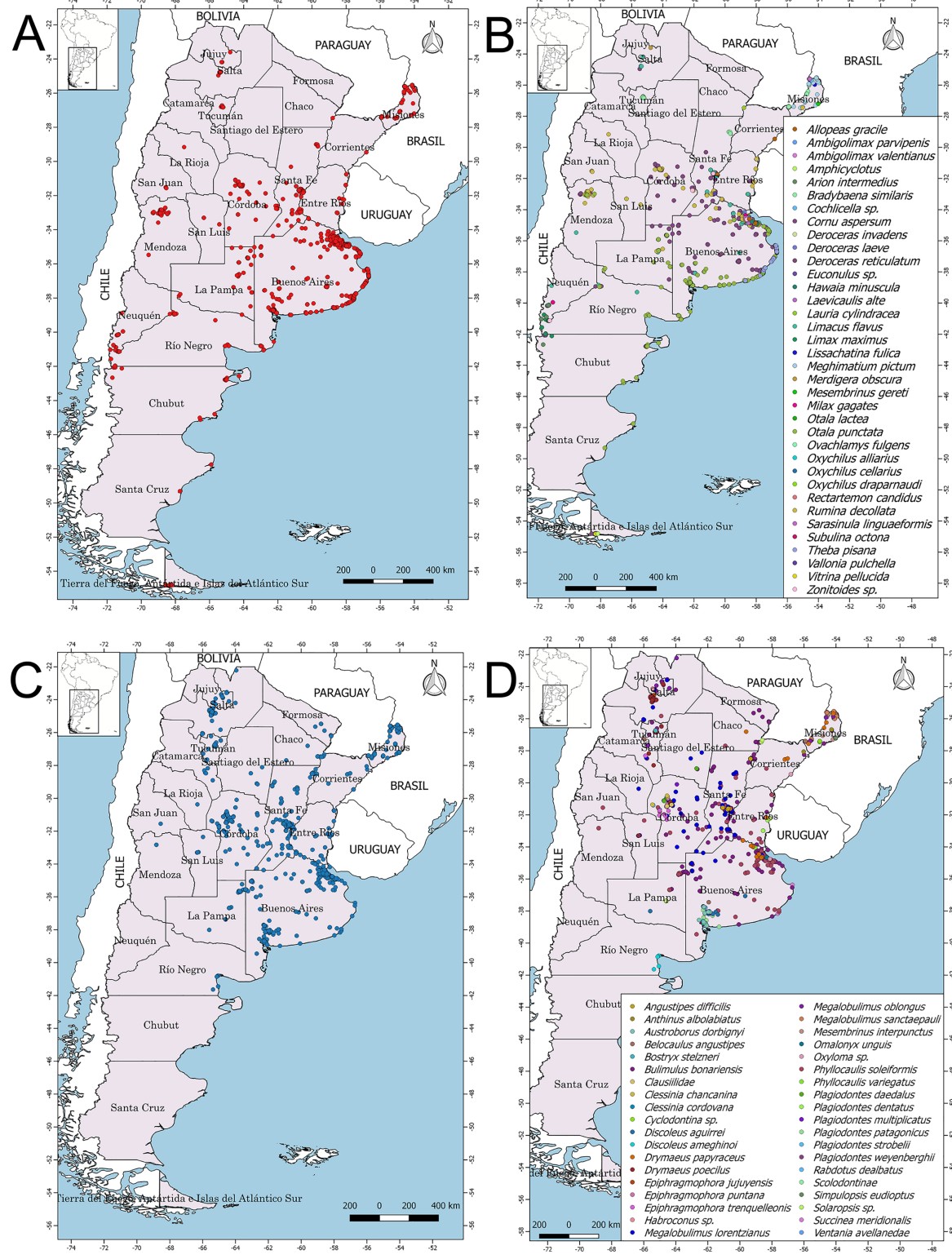

**Figure 2 Geographical distribution of terrestrial gastropods registered in iNaturalist for Argentina.** The general geographic distribution of records for exotic species is shown in red (A) and for native species in light blue (C). (B–D) The geographic distribution of each recorded species, exotic and native, respectively. Image credit: Díaz Ana Carolina.

compared to the geographic distribution known from the literature. In addition, observations of possible species not yet described for science, records that need to be studied because they represent entries of species not yet cited in Argentina, and the records of a species with a live animal were considered (Data Set 2). Of interest for this work were 374 observations (Table 1), representing 9.97% of the total number of observations analyzed, some examples are shown in Figs. 3A–3H.

Regarding the records of land snails for Argentina, the vast majority belong to the Eupulmonata clade, there was only one observation of operculate Cyclophoroidea and no record of Helicinoidea, unlike the records obtained for Brazil (*Rosa, Cavallari & Salvador, 2022*).

Of the exotic species that were recorded on the iNaturalist platform, two species were discarded. *Subulina octona* (Bruguière, 1789) has no cited literature for the country, and the record identified as such (observation 215037922) belongs to a juvenile of *Rumina decollata*. Furthermore, the observation identified as *Ovachlamys fulgens* (Gude, 1900) (163015776) is not consistent with this species. However, *O. fulgens* is recorded for the province of Misiones (*Beltramino et al., 2018*).

As for exotic species, the geographical distribution of records was concentrated in urban areas located in the capitals of the following provinces: Mendoza, Jujuy, Salta, Tucumán, Tierra del Fuego. There have also been many observations of exotic species in the important tourist areas of the provinces of Neuquén and Río Negro. Only some provinces had more extensive records in their territory, such as Buenos Aires, Córdoba, Santa Fe, Misiones, Entre Ríos, and La Pampa. With regard to native species, observations were mainly concentrated in a few provinces such as Buenos Aires, Santa Fe, Entre Ríos, Córdoba, Misiones, Tucumán, Jujuy, and Salta. Other provinces had fewer than ten observations, and there were a few that only had between one and five, as well as others that had no record at all. Thus, the distribution of observations at the national level was uneven and concentrated in a few provinces. In this way, it was found that the highest number of observations is in association with anthropized places, such as provincial capitals and tourist centers.

## DISCUSSION

Citizen science can help detect the spread of exotic species because they are generally synanthropic and prefer disturbed environments. In most cases, species enter, colonize, and spread long before anyone observes the process (*Hutchinson, Reise & Robinson, 2014*). Furthermore, observation, tracking, and monitoring studies are rare. For this reason, user input is very important to accumulate a large number of useful data sources. As in the case of the observation 213840831 (Fig. 3A) identified as *C. aspersum*, based on the appearance of the shell, the coloration of the soft parts, and the large development of the foot, it is very likely that it actually corresponds to the species *Helix pomatia Linnaeus, 1758*, which has not yet been cited for Argentina but has been cited for Brazil (*Salvador et al., 2024*). Other observations that do not correspond to *C. aspersum* and that require a study of their soft parts for a precise identification, and that probably also correspond to species not yet scientifically recorded in Argentina, such as the observations in iNaturalis number

**Table 1 Summary of interesting observations ($n$ = 374) of terrestrial gastropods from Argentina in iNaturalist as of June 15, 2024.** Range extension for the province mentioned (refers to records that extend the known range for the species in the literature); within known range (refers to records that are within the known range); potential new species (corresponds to specimens unknown to science).

| Family | Identification at iNaturalist | Note | Number of selected observations |
|---|---|---|---|
| Helicidae | *Cornu aspersum* | interaction with another species | 1 |
| | *Cornu aspersum* | appears to be *Helix pomatia* | 1 |
| | *Cornu aspersum* | possibly a different species | 7 |
| | *Otala punctata* | range extension (Chubut) | 15 |
| | *Otala punctata* | range extension (Río Negro) | 8 |
| | *Otala punctata* | range extension (Santa Cruz) | 2 |
| | *Otala punctata* | range extension (Mendoza) | 3 |
| | *Otala punctata* | range extension (La Pampa) | 5 |
| | *Otala punctata* | range extension (Neuquén) | 1 |
| | *Otala lactea* | range extension (Mendoza) | 1 |
| Achatinidae | *Rumina decollata* | range extension (Santa Fé) | 7 |
| | *Rumina decollata* | range extension (San Juan) | 3 |
| | *Rumina decollata* | range extension (San Luis) | 7 |
| | *Rumina decollata* | range extension (Entre Rios) | 4 |
| | *Allopeas gracile* | within known range (Buenos Aires) | 2 |
| | *Allopeas gracile* | within known range (Corrientes) | 1 |
| | *Allopeas gracile* | within known range (Entre Ríos) | 1 |
| Limacidae | *Limacus flavus* | range extension (Mendoza) | 3 |
| | *Limacus flavus* | range extension (Santa Fé) | 29 |
| | *Limacus flavus* | range extension (La Pampa) | 15 |
| | *Limacus flavus* | range extension (Salta) | 3 |
| | *Limax maximus* | range extension (Chubut) | 1 |
| | *Limax maximus* | range extension (Buenos Aires) | 3 |
| | *Ambigolimax parvipenis* | collection required for precise identification | 1 |
| Agriolimacidae | *Deroceras reticulatum* | range extension (Neuquén) | 4 |
| | *Deroceras reticulatum* | range extension (La Pampa) | 1 |
| | *Deroceras invadens* | range extension (Buenos Aires) | 3 |
| | *Deroceras laeve* | range extension (Santa Fé) | 1 |
| Milacidae | *Milax gagates* | range extension (Neuquén) | 1 |
| Arionidae | *Arion intermedius* | range extension (Neuquén) | 3 |
| | *Arion intermedius* | range extension (Tierra del Fuego) | 1 |
| Camaenidae | *Bradybaena similaris* | range extension (Santa Fé) | 2 |
| | *Bradybaena similaris* | range extension (Buenos Aires) | 1 |
| Oxychilidae | *Oxychilus cellarius* | collection required for precise identification | 1 |
| | *Oxychilus alliarius* | collection required for precise identification | 1 |
| | *Oxychilus draparnaudi* | cited and registered in Buenos Aires | 2 |
| Gastrodontidae | *Zonitoides* sp. | range extension (Santa Fe) | 1 |
| | *Zonitoides* sp. | range extension (Río Negro) | 1 |
| Pristilomatidae | *Hawaiia minuscula* | within known range (Buenos Aires) | 2 |
| Veronicellidae | *Sarasinula linguaeformis* | within known range (Misiones) | 3 |

(Continued)

| Family | Identification at iNaturalist | Note | Number of selected observations |
|---|---|---|---|
| | *Phyllocaulis soleiformis* | range extension (San Luis) | 2 |
| | *Phyllocaulis soleiformis* | range extension (La Pampa) | 6 |
| | *Phyllocaulis variegatus* | within known range (Misiones) | 13 |
| | *Phyllocaulis variegatus* | within known range (Buenos Aires) | 1 |
| | *Phyllocaulis variegatus* | range extension (Córdoba) | 1 |
| | *Belocaulus angustipes* | possible new location (Misiones) | 1 |
| | *Angustipes difficilis* | within known range (Santa Fe) | 3 |
| | *Laevicaulis alte* | possible first registration (Buenos Aires) | 2 |
| Enidae | *Merdigera obscura* | collection required for precise identification | 1 |
| Geomitridae | *Cochlicella* sp. | collection required for precise identification | 1 |
| Euconulidae | *Euconulus* sp. | collection required for precise identification | 1 |
| | *Habroconus* sp. | within known range (Jujuy) | 1 |
| Valloniidae | *Vallonia pulchella* | range extension (Santa Fe) | 1 |
| Lauriidae | *Lauria cylindracea* | collection required for precise identification | 1 |
| Vitrinidae | *Vitrina pellucida* | within known range (Tierra del Fuego) | 1 |
| Streptaxidae | *Rectartemon candidus* | collection is required to contribute to the knowledge of its distribution. | 1 |
| Neocyclotidae | *Amphicyclotus* sp. | collection required for precise identification | 1 |
| Bulimulidae | *Mesembrinus gereti* | possible new registration for Argentina | 1 |
| | *Bulimulus bonariensis* | potential new species (Santa Fe) | 1 |
| | *Bulimulus bonariensis* | potential new species (Buenos Aires) | 4 |
| | *Bulimulus bonariensis* | potential new species (Formosa) | 1 |
| | *Bulimulus bonariensis* | potential new species (Córdoba) | 1 |
| | *Bulimulus bonariensis* | potential new species (Entre Ríos) | 1 |
| | *Bulimulus bonariensis* | potential new species (Tucumán) | 1 |
| | *Drymaeus poecilus* | range extension (San Luis) | 1 |
| | *Drymaeus papyraceus* | range extension (Chaco) | 1 |
| | *Drymaeus papyraceus* | range extension (Santa Fe) | 1 |
| | *Mesembrinus interpunctus* | range extension (Buenos Aires) | 1 |
| | *Rabdotus dealbatus* | possible erroneous identification | 1 |
| | *Bostryx stelzneri* | within known range (Tucumán) | 1 |
| Bothriembryontidae | *Discoleus aguirrei* | within known range (Buenos Aires) | 10 |
| | *Discoleus aguirrei* | within known range (La Pampa) | 1 |
| | *Discoleus ameghinoi* | within known range (Río Negro) | 3 |
| | *Discoleus ameghinoi* | incorrect identification | 5 |
| Solaropsidae | *Solaropsis* sp. | collection required for precise identification | 2 |
| Clausiliidae | *Clausiliidae* | collection required for precise identification | 2 |
| Cyclodontinidae | *Plagiodontes daedaleus* | within known range (Córdoba) | 9 |
| | *Plagiodontes weyenberghii* | within known range (Córdoba) | 2 |
| | *Plagiodontes patagonicus* | within known range (Buenos Aires) | 19 |
| | *Plagiodontes multiplicatus* | within known range (Córdoba) | 1 |
| | *Plagiodontes strobelii* | within known range (Córdoba) | 6 |

| Family | Identification at iNaturalist | Note | Number of selected observations |
|---|---|---|---|
| | *Plagiodontes dentatus* | within known range (Entre Ríos) | 1 |
| | *Ventania avellanedae* | within known range (Buenos Aires) | 3 |
| | *Clessinia chancanina* | within known range (Córdoba) | 3 |
| | *Clessinia cordovana* | within known range (Córdoba) | 1 |
| | *Cyclodontina* sp. | collection required for precise identification | 1 |
| Strophocheilidae | *Megalobulimus lorentzianus* | range extension (Santa Fe) | 32 |
| | *Megalobulimus oblongus* | range extension (Misiones) | 3 |
| | *Megalobulimus sanctaepauli* | within known range (Misiones) | 13 |
| | *Austroborus dorbignyi* | within known range (Buenos Aires) | 13 |
| Odontostomidae | *Anthinus albolabiatus* | within known range (Corrientes) | 1 |
| | *Anthinus albolabiatus* | within known range (Misiones) | 1 |
| Simpulopsidae | *Simpulopsis eudioptus* | within known range (Misiones) | 2 |
| Scolodontidae | *Scolodontinae* | collection required for precise identification | 1 |
| Epiphragmophoridae | *Epiphragmophora jujuyensis* | within known range (Salta) | 1 |
| | *Epiphragmophora puntana* | potential new species | 1 |
| | *Epiphragmophora puntana* | seems to correspond to another species of *Epiphragmophora* | 1 |
| | *Epiphragmophora puntana* | first image of the living animal | 2 |
| | *Epiphragmophora trenquelleonis* | within known range (Córdoba) | 9 |
| | *Epiphragmophora trenquelleonis* | range extension (Catamarca) | 1 |
| Succineidae | *Oxyloma* sp. | collection required for precise identification | 1 |
| | *Succinea meridionalis* | within known range (La Pampa) | 9 |
| | *Succinea meridionalis* | within known range (Buenos Aires) | 2 |
| | *Omalonyx unguis* | within known range (Santa Fe) | 4 |
| | *Omalonyx unguis* | within known range (Buenos Aires) | 1 |

213866386 (Fig. 3B), 213601591, 198927907, 163736884, 163736884, 134950623, 105855271, 85422883.

For most of the exotic species, the current range based on iNaturalist observations was found to be larger than what was previously known and published. This demonstrates what was mentioned by Hutchinson, Reise & Robinson (2014). Hutchinson, Reise & Robinson (2014) stated that once the presence of the exotic species is known, the population continues to grow and its distribution increases. However, follow-up studies are rarely, if ever, conducted to analyze how the spread of the species into new areas is progressing. In cases such as *R. decollata* and *L. flavus*, the large number of records confirms their presence and wide distribution in Argentina. In addition, these two species in particular are very easy to distinguish simply by analyzing their images, since the species have distinctive characters that allow rapid identification. *Rumina decollata* has a

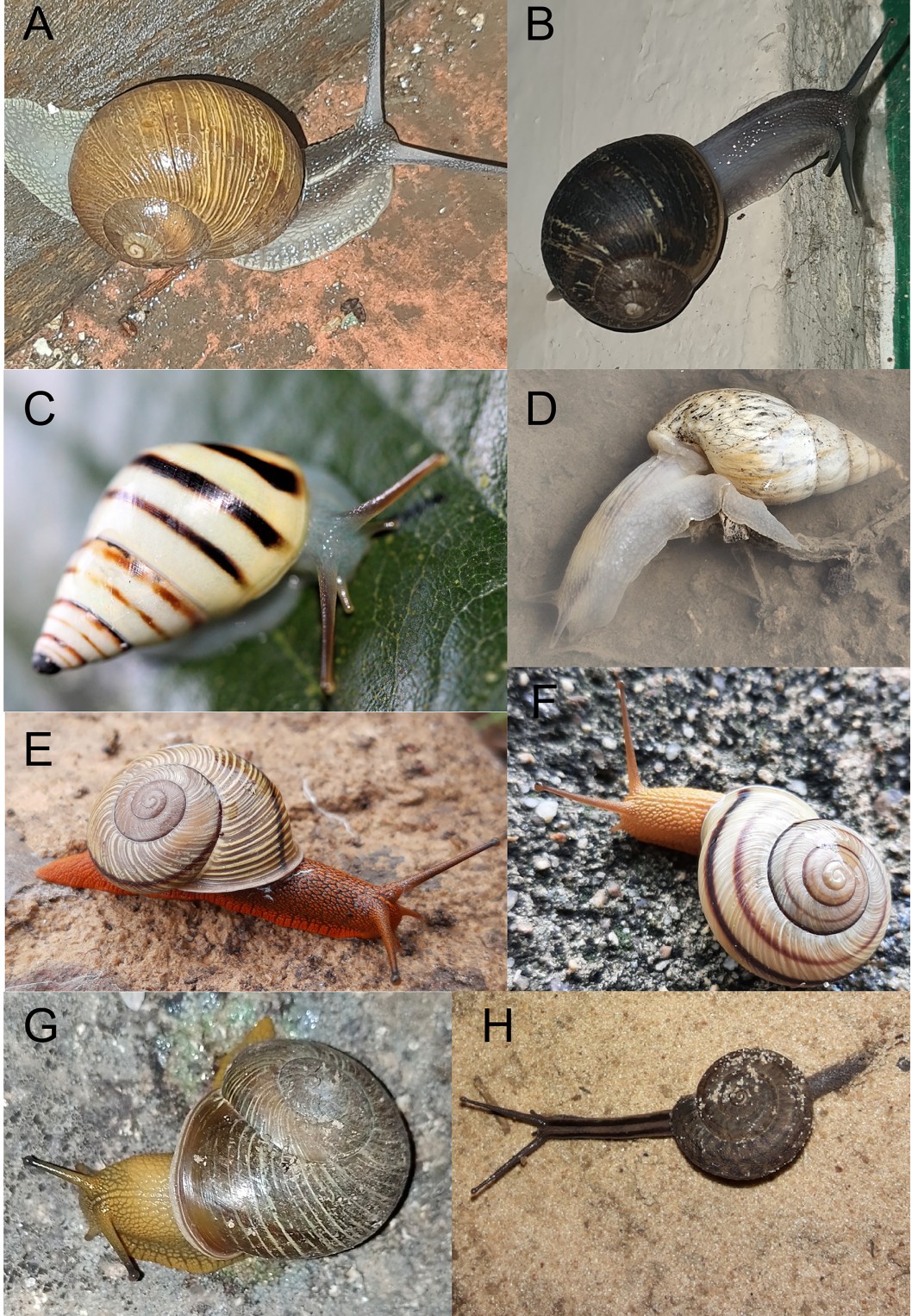

**Figure 3** **Examples of observations of interest on iNaturalist for Argentina as of June 15, 2024.** License is CC BY-NC 4.0 unless otherwise noted. (A) *Cornu aspersum* (observation 213840831, by ignaciostvz (Ignacio Nahuel Cefarelli Estevez), 05/V/2024). (B) *Cornu aspersum* (observation 213866386, by ignaciostvz (Ignacio Nahuel Cefarelli Estevez), 05/V/2024). (C) *Mesembrinus gereti* (observation 148479710, by ricardomoyano (Ricardo Domingo Moyano), 24/V/2017). (D) *Bulimulus bonariensis* (observation 190690046, by fergoracy (María Fernanda Goracy), 11/XI/2023.

**Figure 3** (continued)
(E) *Epiphragmophora puntana* (observation 36379458, by luispiacentini (Luis Piacentini), 22/XI/2019). (F) *Epiphragmophora puntana* (observation 105479501, by nahue (Nahuel Iván Cuba), 23/I/2022). (G) *Epiphragmophora puntana* licencia CC BY (observation 105300293, by tomascarper (Tomás Carranza Perales), 20/I/2022). (H) *Solaropsis* sp. (observation 43632008, by oscar_galli_merino (Oscar Leonardo Galli Merino), 28/V/2019).                               

truncated spire apex (*Linnaeus, 1758*) and its body color varies from light gray with a black middorsal line and yellowish foot to a black body with gray foot (*Prévot, Backeljau & Jordaens, 2015*). *R. decollata* was recently reported by *Rau et al. (2022)* from Buenos Aires, La Pampa, Mendoza, Córdoba, Chubut, Rio Negro, and Misiones. However, through iNaturalist, it is also reported in Santa Fe, San Juan, San Luis and Entre Rios. For its part, *L. flavus* has a "fingerprint-like" shield (*Virgillito, 2012*) and a greenish-gray coloration, with rounded or irregular yellowish spots on the shield, and elongated spots on the rest of the body, for which it is known as the yellow slug. Although *L. flavus* has been reported from Misiones, Tucumán, Corrientes, Córdoba, Buenos Aires, Neuquén, Río Negro and Chubut (*Virgillito, 2012*), iNaturalist has records for Mendoza, Santa Fe, La Pampa and Salta, thus expanding its distribution.

Regarding *O. punctata*, *Virgillito (2012)* cited it in Jujuy, Catamarca, Tucumán, Entre Ríos and Buenos Aires; the records on the iNaturalist platform show a much wider distribution in the Argentine territory, being found in Chubut, Río Negro, Santa Cruz, Mendoza, La Pampa and Neuquén. However, because of its similarity to the species *Otala lactea* (O. F. Müller, 1774), a future morpho-anatomical and genetic revision of the Argentinean populations is necessary. In the literature, *O. lactea* is also cited in several Argentine provinces, such as Santa Fe, Misiones, Tucumán, Córdoba, Entre Ríos, Buenos Aires, Río Negro and Santa Cruz (*Doering, 1874*; *1875*; *Virgillito, 2012*), to which is added the record for Mendoza in iNaturalist.

In the case of the exotic species *T. pisana*, the observations recorded in the iNaturalist platform are consistent with the previously known distribution (*Rumi, Sánchez & Ferrando, 2010*, *Rumi et al., 2019*). There are no records of *T. pisana* outside the province of Buenos Aires. Similarly, no user has yet recorded *L. fulica* and *M. pictum* in localities outside the province of Misiones, which is consistent with recent literature (*Gutiérrez & Beltramino, 2021*; *Sernioti et al., 2021*).

Regarding *B. similaris*, its presence has only been documented in the literature for Tucumán, Misiones and Entre Ríos (*Serniotti et al., 2019*; *Virgillito & Miquel, 2013*). Although a more detailed anatomical and genetic study of the observations made in iNaturalist is required, its presence in Santa Fe and Buenos Aires is possible, since these provinces are neighboring to Entre Ríos. The same is true for *Deroceras laeve* (O. F. Müller, 1774) cited for Buenos Aires and La Pampa (*Costamagna et al., 1999*; *Gutiérrez et al., 2020*; *Reviriego, Descamps & Avila, 1989*); however, a record was found for Santa Fe that needs to be confirmed by anatomical studies. A similar issue was observed for the records of *D. reticulatum*, as it is cited in the literature for the provinces of Tucumán, Buenos Aires, Neuquén, Río Negro, Chubut and Tierra del Fuego (*Fernández, 1973*; *Virgillito & Miquel, 2013*), and in iNaturalist it was found in Neuquén and La Pampa. The same was observed

for the slug *L. maximus*, which was cited for the provinces of Neuquén and Río Negro (*Fernández, 1973*; *Virgillito & Miquel, 2013*) and found by users in Chubut and Buenos Aires. *Deroceras invadens* Reise, Hutchinson, Schunack & Schlitt, 2011 also cited in Argentina for Neuquén, Río Negro and Chubut (*Gutiérrez et al., 2013*), was recorded on the iNaturalist platform in Buenos Aires. The same is true for *Arion intermedius* Normand, 1852, recorded on iNaturalist for Neuquén and Tierra del Fuego; *A. intermedius* was cited by *Gutiérrez et al. (2013)* and *Virgillito (2012)* only for Río Negro and Chubut. In the specific case of *Milax gagates* (Draparnaud, 1801), the species is characterized by a keel in the dorsal region that runs along the animal from the shield to the caudal end (*Virgillito, 2012*), making it easy to distinguish. In addition, the known distribution is for Buenos Aires and Río Negro (*Gutiérrez et al., 2020*; *Virgillito, 2012*), in the analyzed platform it was recorded in Neuquén, due to the proximity between the provinces it is possible that it has expanded its range. This highlights the importance of the use and registration (creation of an account) of users from the community in general on the platform, as they contribute to the monitoring and generation of an early warning of the spread of exotic species. Oxychilidae such as *Oxychilus cellarius* (O. F. Müller, 1774) and *O. alliarius* (J. S. Miller, 1822) have no bibliographic records where they are cited in Argentina, in addition to the fact that the observations in iNaturalist are not post-validation, *i.e.*, they have not achieved research grade, and the identification based on the images is difficult due to the quality of the photographs. In the case of *O. draparnaudi* (H. Beck, 1837), it was cited in Buenos Aires by *Virgillito & Miquel (2013)*, but the contribution of records from iNaturalist and the collection of material in other localities allowed us to expand the knowledge of the distribution range of the species (*Díaz, Martin & Cao, 2025*).

Regarding the representatives of the genus *Zonitoides* Lehmann, 1862, *Virgillito (2012)* indicates that the species *Zonitoides arboreus* (Say, 1817) is widespread in Argentina (Jujuy, Salta, Catamarca, Tucumán, Misiones, Corrientes, Santiago del Estero, Córdoba, Entre Ríos, Buenos Aires). Although the observations of *Zonitoides* made on the platform for Santa Fe and Río Negro have not yet been post-validated, their presence must be confirmed by analyzing the specimens, since it is difficult to determine their presence from their image. The same applies to *Merdigera obscura* (O. F. Müller, 1774) (observation 209964103), *Cochlicella* A. Férussac, 1821 (observation 140362256), *Euconulus* Reinhardt, 1883 (observation 134796118), *Lauria cylindracea* (Da Costa, 1778) (observation 87159107), *Ambigolimax parvipenis* Hutchinson, Reise & Schlitt, 2022 (observation 202705570), *Amphicyclotus* Crosse & P. Fischer, 1879 (observation 137231266). These species have not yet been formally cited for Argentina, and the records in iNaturalist do not have post-validation, so they require the collection of material and detailed studies to determine their identification. A different case occurs with *Laevicaulis alte* (A. Férussac, 1822), where it was observed that the only image (observation 142035812) and with research grade, does not correspond to *L. alte*, because it lacks the clear line running through the center of the mantle (*Brodie & Barker, 2012*). However, two other images from Buenos Aires (observations 92670155 and 38506963), which lack post-validation, seem to correspond to this species because they show this feature. The lack of information in the country alerts us to the need to confirm their identification by collecting and analyzing

material. These data could contribute to the first record of the species in Argentina. Its importance lies in that *L. alte* is considered an agricultural pest of tomatoes, spinach, cucumbers and is of medical interest as a vector of nematodes (*Brodie & Barker, 2012*).

The known records of *Vallonia pulchella* (O. F. Müller, 1774) in Argentina are for the provinces of Jujuy, Buenos Aires, La Rioja and Salta (*Rumi, Sánchez & Ferrando, 2010*); the observation 191307384 made for Santa Fe could contribute to the knowledge of the current dispersion in this country.

With respect to *Rectartemon candidus* (Spix, 1827), although other species of the same genus have been cited in Argentina (*Gutiérrez et al., 2013*), there are no records for this particular species. The shells have few characters of truly reliable taxonomic value and are variable within the group, even at the species level (*Zanin, 2024*). The observation 89826939 for Corrientes could contribute to the knowledge of its distribution, since it is present in Brazil (*Salvador et al., 2024*). However, more detailed studies are needed.

Species that were recorded in iNaturalist within the known ranges of extent in the literature were *Allopeas gracile* (T. Hutton, 1834) (*Virgillito, 2012*), *Hawaiia minuscula* (A. Binney, 1841) (*Miquel, Tablado & Sodor, 2007*; *Rumi, Sánchez & Ferrando, 2010*; *Virgillito & Miquel, 2013*), *Sarasinula linguaeformis* (C. Semper, 1885) (*Santin & Miquel, 2015*), and *Vitrina pellucida* (O. F. Müller, 1774) (*Cuezzo & Dellagnola, 2024*).

On the other hand, *Mesembrinus gereti* (Ancey, 1901) is a Bulimulidae native to Brazil with an interrupted distribution between records from Trinidad and Tobago and Venezuela, with the southernmost records from Minas Gerais (*Macedo, Ovando & D'ávila, 2023*; *Salvador et al., 2023*). The observation 148479710 of *M. gereti* for Misiones could be the first record in Argentina (Fig. 3C).

It was observed that the most records of exotic species correspond to urban, disturbed areas and provinces with high demography, as mentioned by *Rosa et al. (2022)* for *O. fulgens* in Brazil. The main pathway for the spread of exotic species is through international trade (*Barker, 2002*; *Robinson, 1999*). Given the frequency with which exotic species have been found in nurseries and garden centers, colonization of new sites by garden plants is a common event (*Hutchinson, Reise & Robinson, 2014*). In recent decades, gastropods have become increasingly important as crop pests, in some cases becoming serious pests (*Barker, 2002*; *Raut & Barker, 2002*; *Robinson, 1999*). Thus, on a regional level, commercial nurseries can cause the dispersal of terrestrial snails and slugs (native and non-native) through the national and international plant trade, as well as undetected terrestrial transport of eggs and newborn individuals (*Gutiérrez et al., 2020*).

As noted by *Rosa, Cavallari & Salvador (2022)*, the status of the records mentioned so far demonstrates that this platform, as well as others with a focus on citizen science, play a very important role in species monitoring.

In terms of native species, the superfamily Orthalicoidea represents almost 50% of the native fauna of Ecuador (*Breure & Borrero, 2008*). The authors expect similar rates in other South American countries, making representatives of this superfamily an important component of the land snail fauna of this continent. The observations in the iNaturalist platform have different types of biases such as geographical origins concentrated in certain provinces or regions, greater number of records related to macromollusks, low

representativeness of the local fauna due to biases that favor large and synanthropic species, as mentioned by *Rosa, Cavallari & Salvador (2022)*, among others. Considering the issues, the number of species records for Orthalicoidea (19) over the total number of species (38) represents 50% of the native terrestrial mollusks, as mentioned by *Breure & Borrero (2008)*.

Among the observations made on the iNaturalist platform, *B. bonariensis* was the species with the most records (612 with pre-validation and 503 with research grade). *B. bonariensis* is widely distributed in Argentina (Buenos Aires, Entre Ríos, Corrientes, Santa Fe, Formosa, Misiones, Córdoba, Chaco, Jujuy, Salta, Santiago del Estero and Tucumán) (*Miquel, 1991*). When analyzing the records, it was found that the observations 196730052, 190690046 (Fig. 3D), 186089211, 150745443, 150745443, 117795783, 48124517, 39382182, 105358396 could be of scientific interest as potential new species, due to the coloration of both their soft parts and their shell, since they do not seem to correspond to *B. bonariensis* by simple examination of the image.

In the case of native species, data from records in the iNaturalist platform can contribute to the knowledge of their distribution. *P. soleiformis* has a wide distribution (Buenos Aires, Misiones, Jujuy, Salta, Tucumán, La Rioja, Catamarca, Chaco, Santiago del Estero, Corrientes, Córdoba, Mendoza, Santa Fe, Entre Ríos) (*Santin & Miquel, 2015*). For this reason, it was possible that *P. soleiformis* was also present in other provinces. Observations on the iNaturalist platform allowed the species to be recorded in San Luis and La Pampa. *D. poecilus* is reported for Catamarca, Chaco, Córdoba, Formosa, Jujuy, Misiones, Salta, San Juan, Santiago del Estero and Tucumán (*Miquel, 1989*; *Díaz, 2022*), its distribution is so wide that it is also distributed in Bolivia (*d' Orbigny, 1835*), Brazil (*Salvador et al., 2024*) and there is an additional record from San Luis in Argentina. *D. papyraceus* has been recorded in the literature for Buenos Aires, Corrientes, Entre Ríos, Formosa and Misiones (*Miquel, 1989*; *Cuezzo, Miranda & Ovando, 2013*). However, in iNaturalist *D. papyraceus* was recorded for Chaco and Santa Fe, with one record for each province. Due to the proximity of the provinces, it is possible that the distribution of this species is wider than previously known.

For *M. interpunctus* the known distribution is Brazil, Paraguay, and Argentina, where it is restricted to Misiones (*Cuezzo, Miranda & Ovando, 2013*). The only record on the platform (observation 153080583) for Buenos Aires could correspond to a case of accidental translocation or transfer.

For *M. lorentzianus* the known distribution is Salta, Jujuy, Tucumán, Catamarca, Santiago del Estero, Córdoba, Chaco (*Beltramino, 2014*). *Beltramino (2014)* predicts the potential distribution of the species based on a bioclimatic model; the records in iNaturalist for Santa Fe and one for La Rioja agree with this prediction. However, *M. lorentzianus* has also been recorded in Buenos Aires and Entre Ríos. *M. oblongus* is a widely distributed species in Bolivia, Brazil, Colombia, Venezuela, Peru and Argentina, where it is known to occur in the provinces of Jujuy, Salta, Catamarca, Córdoba, Tucumán and in Yunga's environments (*Salas Oroño, 2018*). The records in iNaturalist for Misiones are far from its known distribution. Based on the evidence compiled by *Beltramino (2016)*, *M. oblongus* would be restricted to only a small area in northwestern Argentina and therefore would not

contribute to the terrestrial malacofauna of Misiones province. Correct identification will require studies of the ultrastructure of the protoconch (ribs and microgranulation), the shape of the shell and spire, as well as anatomical and genetic studies.

For native slug species, appearance and colouration can help with identification, but more detailed studies are often required. *P. variegatus* is distributed in Misiones, Entre Ríos and Buenos Aires (*Santin & Miquel, 2015*), and most of the records are consistent with the distribution data in the literature. The species is characterized by having a brown notum with two longitudinal lines of dark spots that meet at the caudal end, and the observation 194356837 agrees with this description. However, in ventral view it is light brown with a narrow foot and *P. variegatus* is characterized by a wide foot (*Santin & Miquel, 2015*). Therefore, material from this province requires analysis of its anatomy to rule out or confirm the presence of the species.

For native species, we also found records with distributions in iNaturalist that were consistent with the literature. *M. sanctaepauli* known for Brazil, Paraguay, and Argentina (Misiones and Corrientes) (*Beltramino, 2012*) with observations in iNaturalist for Misiones; *Clessinia chancanina* (Doering, 1878) cited for the Sierras de Pocho (Chancani) in Córdoba (*Hylton Scott, 1965*), recorded for the same location and in nearby sierras. *P. patagonicus* cited and recorded for Buenos Aires (*Pizá & Cazzaniga, 2003*); as well as *Plagiodontes weyenberghii* (Doering, 1877) for Córdoba (*Gordillo et al., 2013*), and *Clessinia cordovana* (Pfeiffer, 1855) for the same province (*Cuezzo, de Lima & Dos Santos, 2018a*); *Plagiodontes multiplicatus Doering, 1874*, for central-western Argentina (*Pizá & Cazzaniga, 2003*). *Bostryx stelzneri* (Dohrn, 1875) belongs to a complex of species with a wide distribution ranging from Bolivia to San Luis (Argentina) (*Miranda & Cuezzo, 2014*), with a record for Tucumán. *Anthinus albolabiatus* (Jaeckel, 1927) was cited and with records in Argentina (Misiones and Corrientes) (*Bonard, Caldini & Miquel, 2012*) also in Paraguay (*Quintana, 1982*), and Uruguay (*Klappenbach & Olazarri, 1984*). *Simpulopsis eudioptus* (Ihering, 1897) has also been observed in Misiones, one of the two provinces where it is distributed together with Jujuy (*Cuezzo, Miranda & Ovando, 2013*). The record 19949188 identified as *Habroconus* Crosse & P. Fischer, 1872 for Jujuy corresponds to the known distribution for the genus. *Miranda & Cuezzo (2010)* found specimens of *Habroconus* sp. in the Sierra de San Javier Biological Park, Tucumán. The park has characteristics of the Yungas, which extend from the border with Bolivia to Argentina (north of Catamarca, west of Salta, Jujuy and Tucumán), so the observation is within this area.

*Epiphragmophora* is a typical component of the Andean fauna, distributed in Peru, Bolivia, Paraguay, southern Colombia, southern Brazil and Argentina from high altitudes to the plains (*Cuezzo, 2006*). Most of the species in the genus are known by their shell morphology, and for the Argentinean species *Fernández & Rumi (1984)* and *Cuezzo (2006)* studied their anatomy. Although *Cuezzo (2006)* described the appearance of the body, she did not include pictures of living specimens of the species. *Epiphragmophora puntana* (Holmberg) inhabits San Luis, Córdoba, La Rioja and Tucumán (*Fernández & Rumi, 1984*), in iNaturalist it was only recorded in Córdoba with images (36379458, 22227718), where the reddish to orange body colour can be seen as described by *Cuezzo (2006)*.

Nevertheless, these observations are the first photographic records of the living animal (Fig. 3E). Another observation (105479501) was identified as *E. puntana* based on the general appearance of the shell with three dark brown bands, not very thin bands separated by wider spaces, a surface with distinct growth striations and with a slightly elevated spire (*Fernández & Rumi, 1984*). Furthermore, it seems to correspond to *Epiphragmophora trigrammephora* (d'Orbigny, 1835) due to its distribution in the northwestern and central region of Córdoba (*Cuezzo, 2006*), it seems to correspond to *Epiphragmophora trigrammephora* (d'Orbigny, 1835). On the other hand, *Epiphragmophora guevarai Cuezzo, 2006* shows similar characteristics in the shell and they coincide in their distribution (*Cuezzo, 2006*). The differences between the two are in the size of the adult shell, which is smaller in *E. guevarai*, and in their genital structures. Thus, although an analysis of its anatomy is necessary, this is the first record in the living animal for the species it turns out to be (Fig. 3F). This observation highlights the importance of adding an object to the images that indicates the size of the animal photographed and thus helps in its identification. On the other hand, the observation 105300293, identified as *E. puntana*, has a shell that does not seem to correspond to the other species of *Epiphragmophora* found in Córdoba and could be a possible new species (Fig. 3G). *Epiphragmophora jujuyensis* Hylton Scott, 1962 is a typical inhabitant of the cloud forest or Yungas, although it is not easy to find, it is distributed in Salta and Jujuy (*Cuezzo, 2006*; *Fernández & Rumi, 1984*), shell remains have been photographed in Salta. *E. trenquelleonis* is recorded for Córdoba, Santiago del Estero, San Luis, Catamarca, La Rioja, Chaco and Formosa (*Cuezzo, 2006*), but only for Córdoba and Catamarca on the platform.

*Plagiodontes dentatus* (Wood, 1828) with records in Entre Ríos and cited for that province, northern Buenos Aires and Uruguay (*Pizá & Cazzaniga, 2003*); *Ventania avellanedae* (Doering, 1881) cited and found in Buenos Aires (*Pizá, Cazzaniga & Ghezzi, 2018*); *Discoleus aguirrei* (Doering, 1884) cited for Buenos Aires and La Pampa (*Cuezzo, Miranda & Ovando, 2013*) and recorded in iNaturalist for both provinces; *Austroborus dorbignyi* (Doering, 1876) cited in the literature for Sierras de la Ventana and Sierras de Curamalal, southern Buenos Aires Province (*Bonard, Caldini & Miquel, 2012*) and with photographic observations south of that province in sierras and around towns. *Succinea meridionalis* d'Orbigny, 1846 is also widely distributed in Peru, Brazil, Bolivia, Paraguay, Uruguay and Argentina, in tropical and temperate areas to the north of Patagonia (*Hylton Scott, 1963*; *Miquel & Aguirre, 2011*), but has only been recorded from La Pampa and Buenos Aires. *Plagiodontes daedaleus* (Deshayes in Férussac & Deshayes, 1820) is mentioned for Córdoba in the sierra region and extends to other provinces around the Sierras Pampeanas (*Boretto et al., 2015*), occupying central-western Argentina (*Miquel & Aguirre, 2011*). On the iNaturalist platform, *P. daedalus* has been recorded for Córdoba and Santiago del Estero. *Plagiodontes strobelii* (Doering, 1877) is cited for Córdoba (*Gordillo et al., 2013*) and is the only province from which records were obtained. *Omalonyx unguis* (A. d'Orbigny, 1836) is cited for Buenos Aires, Santa Fe, Formosa, Chaco (*Coscarelli & Vidigal, 2011*), and Misiones (*Guzmán et al., 2018*), with records in iNaturalist only for Santa Fe and Buenos Aires.

*Belocaulus angustipes* (Heynemann, 1885) is native to southern South America and occurs in Uruguay, Paraguay, Brazil, and in Argentina in Santa Fe, Tucumán and Buenos Aires (*Gutiérrez et al., 2020*; *Ohlweiler, Mota & Gomez, 2009*). In iNaturalist, *B. angustipes* has been recorded in the latter two provinces mentioned and presents a record without post-validation for Misiones. The observation 30041592 matches the description in external appearance, so it is known as the black velvet slug. However, as in other cases, the confirmation of belonging to this species is only based on the anatomy of the male reproductive system (*Gomes, 2007*).

In the case of the slug *Angustipes difficilis* (Colosi, 1921), which also has wide distribution (Salta, Tucumán, Misiones, Chaco, Corrientes, Santa Fe, Entre Ríos and Buenos Aires (*Santin & Miquel, 2015*)), there were only records for Santa Fe.

For most of these species, the observations recorded on the iNaturalist platform are in a few provinces relative to a broader actual distribution. Reading this data shows that although citizen science gives us access to a lot of data to analyze and discuss, it is still not a widespread and widely used tool among the population. As in other countries, we hope that its use will become more widespread over time and with an eye to the future. In this way, citizen science will allow us to make different interpretations and contribute to the knowledge of both native and exotic fauna, in this case for Argentina, so that we can achieve positive results for science and society.

We have also found records where the identification is clearly incorrect, as they correspond to another species, as in the case of *Discoleus ameghinoi* (Ihering, 1908). *D. ameghinoi* is characterized by a beige oval-conical shell with longitudinal brown stripes. Its distribution is Buenos Aires, Río Negro, La Pampa and Santa Cruz (*Cuezzo, Miranda & Ovando, 2013*; *Fernandez, 1969*), and although there are records that correspond to this distribution, some of the images do not. A similar case is the observation 222651057, identified as *Rabdotus dealbatus* (Say, 1821). This record shows the great similarity between the members of the family Bulimulidae, whose distribution in Argentina is wide and can easily be confused, which is why it is necessary to carry out more in-depth studies. It is because of these identification errors that some people question the practice of citizen science, questioning the quality of the data. Therefore, it is necessary to develop appropriate protocols, provide training and supervise volunteers when collecting data that can be used by science (*Bonney et al., 2014*). It also shows that researchers have an important role to play in checking the information that is available and to be used.

On the other hand, there have been observations of specimens, without post-validation, identified as *Solaropsis* sp. The distribution of the genus extends from Costa Rica to northern Argentina, and from Colombia eastward to Suriname and French Guiana. The poor general knowledge of the group and the lack of anatomical and molecular information have been highlighted among researchers (*Cuezzo, de Lima & Dos Santos, 2018a*). Therefore, having data on where they were found or sighted is of great value for future collections, as well as capturing the appearance of the live animal (Fig. 3H).

The observations of Clausiliidae in Argentina were only two (195786275, 156062693), both for Córdoba and without post-validation. *Hausdorf & Neiber (2022)* reported a wide

distribution range from Europe, Africa, South and East Asia, the Greater Antilles, and South America. The species of this family are characterized by the clausillial apparatus, a series of structures on the shell used to close the opening. Like the Cyclodontinidae *Salvador et al. (2023)* they have a series of folds on the walls of the opening and have a wide distribution in South America (*Cuezzo et al., 2018b*). For the reasons mentioned above, we show that the observation of these structures is necessary to ratify the identification of the observations. This comment is also applies to the observation 134987098, which has no post-validation and has been identified on the platform as Cyclodontina.

The record 129050253, identified as belonging to the genus *Oxyloma* sp., has no post-validation and no bibliography has been found where the species is cited in this country, which requires more exhaustive studies. The species *Oxyloma beckeri* Lanzieri, 1966 (*Salvador et al., 2024*) was recorded for Brazil, for which there are no records in Argentina.

The *Scolodonta* comprise about 23 species distributed in temperate and tropical areas of South America including Argentina (*Miquel & Santin, 2020*). The observation 222711592 identified as Scolodontinae seems to belong to a different genus because it has an elevated spire, unlike *Scolodonta* which has an almost absent spire (*Miquel & Santin, 2020*).

## CONCLUSIONS

Throughout this analysis, we have been able to demonstrate the relevance of citizen science in providing interesting contributions to the knowledge of terrestrial mollusks biodiversity in Argentina. Citizen science offers the possibility of having large geo-referenced datasets that are useful for multiple applications. These applications include the analysis of the presence and distribution of species, which is a useful tool for monitoring mainly pest species in agriculture. This has led to new quantitative approaches to the distribution and abundance of organisms at unprecedented temporal and spatial scales (*Dickinson, Zuckerberg & Bonter, 2010*; *Rosa, Cavallari & Salvador, 2022*; *Vendetti et al., 2018*).

This work allowed us to contribute to detect the expansion of the range of exotic species such as *R. decollata*, *L. flavus*, *B. similaris*, *D. laeve*, *D. reticulatum*, *D. invadens*, *A. intermedius*, *M. gagates*, *L. maximus*, *V. pulchella*, possibly *L. alte*, and native species such as *P. soleiformis*, *D. poecilus* and *D. papyraceus*. We also recorded species within the known range, a possible case of accidental transfer for *M. interpunctus* was detected, and the prediction of the distribution of *M. lorentzianus* was checked.

On the other hand, we can mention some other examples to comment on the scope of these community use platforms. *Rosa et al. (2022)* were able to assess the current distribution of *O. fulgens* in Brazil using data from the literature, together with material available in collections, field samples and records in iNaturalist, which showed an alarming expansion towards other states with protected areas. Similarly, *Díaz, Martin & Cao (2025)* were able to assess the current distribution of *O. draparnaudi* in Buenos Aires. This makes iNaturalist a useful tool for monitoring and controlling exotic species. Another interesting application of citizen science is to facilitate the detection of the first appearance of exotic species with pest potential at the site where they become established. Although they do not

have the potential to become pests, this work identified species not recorded in Argentina such as *H. pomatia*, *M. gereti*.

Moreover, it proved to be a great help in finding rare or little known species, such as those mentioned by *Rosa, Cavallari & Salvador (2022)* for members of Simpulopsidae, Streptaxidae and Strophocheilidae, as well as undescribed species unknown to science, such as *Plekocheilus* sp. and *Megalobulimus* sp. (*Rosa, Cavallari & Salvador, 2022*). In this work, we detected potential new species of *Bulimulus* and *Epiphragmophora*. Even *Rosa, Cavallari & Salvador (2022)* found records of living species that were thought to be extinct, such as *Leiostracus carnavalescus* Simone & Salvador, 2016 and *Gonyostomus egregius* (L. Pfeiffer, 1845). Citizen science has also proved valuable in learning about the live appearance of species described only by their shells and whose soft parts are unknown. Several species of *Epiphragmophora* were mentioned in this article, and *Rosa et al. (2022)* were able to find the first records of the live animal for species of Helicinidae, Bulimulidae, Solaropsidae, Strophocheilidae, as well as *Aperostoma amazonense* Bartsch & J. P. E. Morrison, 1942, *Moricandia angulata* (J. A. Wagner, 1827), *Orthalicus phlogerus* (A. d'Orbigny, 1835), *Leiostracus vimineus* (S. Moricand, 1834).

In addition, species that require detailed anatomical studies for precise identification were mentioned throughout the article. Therefore, this review of the records in iNaturalist has uncovered gaps in knowledge about terrestrial gastropods in Argentina.

From another perspective, citizen science is also an important pillar in biodiversity conservation, providing data to help to understand environmental change at large geographical scales. In addition, surveillance and monitoring of priority species allows estimating patterns and trends, and consequently provide data to manage public policies for environmental protection (*Dickinson, Zuckerberg & Bonter, 2010*; *McKinley et al., 2017*). For its part, the Global Biodiversity Information Facility (GBIF) exports "research-grade" iNaturalist records on a weekly basis (*GBIF.org, 2025*). Its goal is to provide free and open access to biodiversity data from around the world to support scientific research, promote biological conservation and foster sustainable development.

These are just a few examples of how citizen science platforms are being used and applied in science. Over time, as its use becomes more widespread, and given the potential it has shown, we believe it could have many other applications in the future that we cannot even imagine today.

Using this and other work as examples, we encourage the scientific community to consult the records on the iNaturalist platform or other citizen science platforms, as they can provide complementary data to help inform their research.

## ACKNOWLEDGEMENTS

We thank the users of the iNaturalist platform for their records of land snails and, especially, those who have authorized the publication of photos of their authorship.

### Funding

Financial support for this work was provided by an institutional project from the Facultad de Ciencias Naturales y Museo, Universidad Nacional de La Plata (Proyect N1026); and Proyect PICT 03585 FONCyT. The funders had no role in study design, data collection and analysis, decision to publish, or preparation of the manuscript.

### Grant Disclosures

The following grant information was disclosed by the authors:
Facultad de Ciencias Naturales y Museo, Universidad Nacional de La Plata: N1026 and PICT 03585 FONCyT.

### Competing Interests

The authors declare that they have no competing interests.

### Author Contributions

- Ana Carolina Díaz conceived and designed the experiments, performed the experiments, analyzed the data, prepared figures and/or tables, authored or reviewed drafts of the article, and approved the final draft.
- Stella M. Martin conceived and designed the experiments, authored or reviewed drafts of the article, and approved the final draft.

### Data Availability

The list of all terrestrial gastropods observations for Argentina, and those of interest for this work, are available in the Supplemental Files.

### Supplemental Information

Supplemental information for this article can be found online at http://dx.doi.org/10.7717/peerj.19152#supplemental-information.

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
