# Peer review of "Use and application of iNaturalist on land snails from Argentina"

_PeerJ, doi:10.7717/peerj.19152_

## Round 0.1 · original submission · Major Revisions

Dear colleagues,

The manuscript has been reviewed by 2 researchers and they both agree that many valuable insights are presented, with which I agree.

They have both made several comments on improving the manuscript, and I highlight:

1. please consider expanding the methods section to include more details on how data was handled. both reviewrs made remarks regarding this aspect;

2. I suggest, as reviewr #2 points out, that separating results from discussion is best. Results should include exclusively data that was generated, while discussion compares more broadly and there is even room for some speculation (as long as it is explicit).

Please consider addressing these issues!

kind regards,

dan

Reviewer 1 ·

Basic reporting

This manuscript provides a very interesting review of the data provided by iNaturalist for land snails in Argentina. While similar research has been done in the past, it serves not only to further support the strengths of iNaturalist as a tool for researchers but also to add to our knowledge on several land snail species mentioned throughout the text.

Overall, the manuscript is well-written, although a few sentences are ambiguous and should be rewritten to be clearer. These sentences are highlighted in the annexed pdf with comments on why and how they should be changed.

The text is organized into three main sections: "Introduction", "Materials & Methods" and "Results and Discussion". The latter section comprises most of the manuscript, encompassing information that usually would be provided in the traditional Results, Discussion and Conclusion sections. Since most of the presented information consists of comparing data from iNaturalist with data from literature, combining the results and their discussion makes some sense. However, I feel like this organization makes it harder for the reader to understand the major points of the manuscript, as it is necessary to read through the entire text to get to the conclusions. I suggest at least separating the final paragraphs of the manuscript into a separate Conclusion section, as I also highlighted in the annexed pdf.

Data presentation through figures, tables and supplementary files is excellent and needs no change.

Experimental design

The research questions of the manuscript are well-defined and relevant to the current research on land snails. It provides a very welcome investigation of the iNaturalist platform as a source of data (supporting previous research on the same topic) and a new investigation on its several records of land snail species in Argentina.

The methods are justified and compatible with previous research conducted on this topic. However, I feel like the Methods section is a bit too short and superficial. Perhaps it could be expanded to provide more details into how the investigation was conducted, such as how the species identification was reviewed and what sources (e.g., literature, collection specimens) were used to determine whether it was correct or incorrect.

Validity of the findings

The research presented is valid, provides all necessary data and has well-supported conclusions.

Additional comments

I congratulate the authors for their research. Studies reviewing and providing basic data, such as distribution data, for the land snail species in South America are severely needed. Your manuscript is a very welcome contribution and hopefully it will help to advance our understanding of the Argentinian biodiversity. Integrating and reviewing citizen science data is also a commendable trend that should be encouraged in future research.

I think the manuscript could be improved with a few minor changes, which I highlight in the annexed pdf. These are mostly related to the language and organization of the text, which I believe will help readers better understand and appreciate your research. Some technical comments on a few species with corrections or suggestions are also provided.

Annotated reviews are not available for download in order to protect the identity of reviewers who chose to remain anonymous.

Reviewer 2 ·

Basic reporting

This article describes the iNaturalist observations of land snails within Brazil and analyzes their utility for more completely understanding this biodiversity.

The English and phrasing, though generally acceptable, is not “clear and unambiguous” throughout. The authors should review again and perhaps use the assistance of colleagues to improve clarity.
Literature references, sufficient field background/context provided.

There is reasonable and appropriate use of background information and citations.

Raw data are shared. The Methods could be revised and expanded, for example, to explain “pre-validation and post-validation”. Did the authors add their own identifications to those on iNaturalist? Existing tables are fine, though table legends should be expanded. Information from the Results and Discussion could be added into a table to make it more clear for the reader. Please see comments made by this reviewer on the manuscript.

Results and Discussion should not be combined; there is no Conclusions section, which might be fine, but it is one of the standard sections for this journal: https://peerj.com/about/author-instructions/

Experimental design

This study has relatively simple experimental design as it is a review of existing data (as iNat observations). It is stated in the article how these data “fill an identified knowledge gap”; however, the Methods are not clear enough to the reader to know how they “cleaned” the data and how they analyzed them using Excel, even as simple percentages. As mentioned above, I would recommend a revised Methods section with more detail.

Validity of the findings

That Results and Discussion are combined makes for some confusion. There is no Conclusions section.

Additional comments

There are lots of comments and suggestions for the author (made by this reviewer in the accompanying annotated manuscript. I hope that they are useful.

Annotated reviews are not available for download in order to protect the identity of reviewers who chose to remain anonymous.

---

## Round 0.2 · accepted · Accept

All comments by reviewers were addressed in a satisfactory manner. I thank the authors for their contribution!

Reviewer 1 ·

Basic reporting

This manuscript provides a very interesting review of iNaturalist data for land snails in Argentina. I commend the authors for their in-depth discussion of all noteworthy observations, greatly adding to our knowledge on Neotropical snails and further supporting the strengths of iNaturalist as a tool for researchers.

Data presentation through figures, tables and supplementary files is excellent. The reorganized text, with the separation between Results, Discussion and Conclusions, is indeed much clearer and easier to read.

Experimental design

The research questions of the manuscript are well-defined and relevant to the current research on land snails. The methods are justified and compatible with previous research conducted on this topic. The revised text in the Methods section is much clearer and improved.

Validity of the findings

The research presented is valid, provides all necessary data and has well-supported conclusions.

Additional comments

I congratulate the authors for their study, which is a very welcome contribution to land snail research and provides a lot of crucial data on the studied species. Hopefully, this manuscript will encourage researchers to continue integrating and reviewing citizen science data in the future.

All my previous comments have been addressed and I think the manuscript is now fit for publication.